# The Association between Dietary Patterns and Pre-Pregnancy BMI with Gestational Weight Gain: The “Born in Shenyang” Cohort

**DOI:** 10.3390/nu14122551

**Published:** 2022-06-20

**Authors:** Jiajin Hu, Ming Gao, Yanan Ma, Ningyu Wan, Yilin Liu, Borui Liu, Lin Li, Yang Yu, Yang Liu, Bohan Liu, Deliang Wen

**Affiliations:** 1Health Sciences Institute, China Medical University, Shenyang 110122, China; jjhu@cmu.edu.cn (J.H.); gaoming@cmu.edu.cn (M.G.); wnyjudy@163.com (N.W.); cmuylliu@163.com (Y.L.); 2021120360@cmu.edu.cn (B.L.); yyu90@cmu.edu.cn (Y.Y.); yliu0568@cmu.edu.cn (Y.L.); 2Department of Epidemiology and Health Statistics, School of Public Health, China Medical University, Shenyang 110122, China; ynma@cmu.edu.cn; 3Development of Pediatrics, Shengjing Hospital of China Medical University, Shenyang 110004, China; lilin@cmu.edu.cn; 4College of Humanities and Social Sciences, Harbin Engineering University, Harbin 150001, China; heu10131107@163.com

**Keywords:** maternal dietary patterns, gestational weight gain, pregnant women, birth cohort study

## Abstract

The reported associations of maternal dietary patterns during pregnancy with gestational weight gain are inconsistent, especially among the less studied Asian Chinese populations. In a prospective pre-birth cohort study conducted in northern China, we determined the associations between maternal dietary patterns and the probability of excess gestational weight gain (EGWG) among 1026 pregnant women. We used 3-day food diaries to assess maternal diet and performed principal component analysis to identify dietary patterns. Maternal adherence to a traditional pattern, which was characterized by a higher intake of tubers, vegetables, fruits, red meat, and rice, was associated with a higher probability of EGWG (quartile 3 vs. quartile 1, odds ratio [OR] = 1.62, 95% confidence interval [CI] = 1.10−2.38). This risk association was more pronounced among women who were overweight/obese before pregnancy (quartile 4 vs. quartile 1, OR = 5.17, 95% CI = 1.45–18.46; *p* for interaction < 0.01). Maternal adherence to a high protein pattern, which was characterized by a higher intake of fried foods, beans and bean products, dairy products, and fruits, was associated with a lower risk of EGWG (quartile 3 vs. quartile 1, OR = 0.56, 95% CI, 0.39−0.81). The protective association was more pronounced among non-overweight/obese women (*p* for interaction < 0.01). These findings may help to develop interventions and better define target populations for EGWG prevention.

## 1. Introduction

The epidemic of excessive gestational weight gain (EGWG) during recent decades has led to a high incidence of adverse maternal and child health outcomes, such as gestational diabetes mellitus, cesarean delivery, and childhood obesity [1,2,3,4]. The prevalence of EGWG in China varies from 27.5−44.5% based on the geographic region [5,6] and has become a serious public health concern.

Emerging evidence suggests that prenatal nutrition has a significant influence on gestational weight status, and thus may be a key modifiable factor in the prevention of EGWG [7,8,9,10,11,12,13,14,15]. Existing studies have examined the role of single nutrients, such as individual macronutrients, and food items on weight gain during pregnancy [8,10,11], but overlooked the complex interaction between foods and nutrients. Maternal dietary patterns represent a broader picture of food consumption during pregnancy and are more relevant and practical for the prevention of EGWG, especially among Asian Chinese populations, who have a complex food composition and diet culture [16,17]. However, only two studies were based on Chinese populations and have reported conflicting results [18,19]. Yang et al. [17] reported that pregnant women following a beans-vegetables dietary pattern had significantly lower gestational weight gain (GWG); however, the bean-vegetable pattern was not associated with GWG in another cohort study in China [18]. Furthermore, both studies [18,19] were conducted in southern China, which may not be representative of the northern food environment in China. Additional studies, therefore, are warranted.

It has been concluded from the existing studies that the associations between maternal calorie, total fat, and cholesterol intake during pregnancies with GWG were influenced by the pre-pregnancy weight status [12,14,20]. These findings indicate that the association between maternal diet during pregnancy on GWG varies according to different pre-pregnancy body mass index (BMI); however, how pre-pregnancy BMI influences the association between maternal dietary patterns with the probability of EGWG is less well studied. Determining the association between pregnant women’s dietary patterns with the probability of EGWG may help to identify the target population for EGWG prevention.

To address these gaps, we examined the associations between prenatal dietary patterns and pre-pregnancy BMI with gestational weight gain, using the data from a pre-birth cohort in northern China.

## 2. Materials and Methods

### 2.1. Study Design

We used data from the Born in Shenyang Cohort Study (BISCS), a prospective pre-birth cohort in the northern area of China. The design of BISCS has been described elsewhere [21]. We recruited women with singleton gestations in the second trimester (22 ± 1.2 weeks gestation) from April to September 2017 in 54 women and children health care institutions in urban areas of Shenyang, Liaoning Province. Among 1296 women with singleton live births, we further excluded pregnancies with incomplete dietary assessments or implausible values of daily calorie intake (<500 or >5000 kcal/day; *n* = 152), implausible values of total GWG (<−5 kg or >40 kg; *n* = 9), and pregnancies in which the final weight was determined earlier than 1 month before delivery (*n* = 109), leaving 1026 participants in the final analysis. 

During the enrollment visit at 22 ± 1.2 gestational weeks, clinical researchers collected pregnant women’s data on their demographic and socio-economic status, medical history, environmental exposure, and personal lifestyles during pregnancy. Researchers performed follow-up evaluations at each antenatal care visit until delivery and collected weight information. All pregnant women provided written informed consent at the enrollment visit and the Ethics Committee of China Medical University approved the study.

### 2.2. Exposures: Maternal Diet

We assessed the maternal diet during pregnancy (22 ± 1.2 weeks gestation) using 3-day food diaries (TFDs). We provided the TFD questionnaires to pregnant women and trained them to record their daily consumption of all food and beverages. Participants were asked to write down all food consumed over three days (including 2 working days and 1 weekend day), with ingredients and portions (in grams). We also provided a visual aid book with photos of local foods in different portions to help the participants better identify the portion size of foods. Pregnant women filled out the TFDs before the oral glucose tolerance test at a mean of 24 weeks gestation (SD 1.2). We summarized all food and beverage items into 21 non-overlapping food groups (Appendix A). According to the Chinese food composition database [22], we calculated the average daily energy and nutrient intakes of pregnant women over 3 days.

### 2.3. Outcomes: Gestational Weight Gain

Participants reported their pre-pregnancy weight (kg) at the time of enrollment and trained obstetricians measured weight during pregnancy a median of 10 times (range, 5–17). The obstetricians measured each participant’s weight twice and calculated the average value of the two measurements. GWG was defined as the difference between the last measured and pre-pregnancy weights. We defined EGWG according to Institute of Medicine (IOM) guidelines, as follows: GWG > 18 kg for pre-pregnancy underweight women; GWG > 16 kg for pre-pregnancy normal-weight women; GWG > 11.5 kg for pre-pregnancy overweight women and GWG > 9 kg for pre-pregnancy obese women [23]. We also calculated the GWG rate (kg/week) using the total GWG divided by gestational age at the time of delivery.

### 2.4. Covariates

We collected participants’ socio-economic and lifestyle information at the time of recruitment using interviewer-administered questionnaires. We calculated women’s pre-pregnancy BMI using the self-reported pre-pregnancy weight (kg) divided by the height (measured) squared (m^2^). We grouped pre-pregnancy weight status into three BMI categories: underweight (<18.5 kg/m^2^); normal-weight (18.5–25.0 kg/m^2^) and overweight/obese ([OwOb] ≥ 25.0 kg/m^2^) according to the WHO references [24]. We categorized participants’ age into four groups (<25, 25–29, 30–34, and ≥35 years), ethnicity into two groups (Han vs. minority), education attainment into two groups (high school or below vs. college or above), annual household income into two groups (<¥50,000/year vs. ≥¥50,000/year), smoking status during pregnancy into two categories (no vs. yes), and parity into two groups (primipara vs. multipara). Prenatal physical activity status was assessed using the Pregnancy Physical Activity Questionnaire (Chinese version) [25], and categorized into three groups (<100 metabolic equivalents [MET]-h/week, 100 to < 200 MET-h/week, and ≥200 MET-h/week). We categorized women’s daily energy intake into two groups (<2100 vs. ≥ 2100 kcal/day) based on the recommended energy intake for the mid-pregnancy women [26]. 

### 2.5. Statistical Analyses

Principal component analysis was used to identify prenatal dietary patterns. Varimax rotation was applied to improve interpretability. We defined distinct dietary patterns according to the factor interpretability (after varimax rotation), eigenvalue, and a scree plot (Appendix A). We calculated dietary pattern scores by summing the standardized food intake level by weighted corresponding factor loadings. According to a previous study [27], we identified the main influencing factor of dietary patterns based on the absolute factor load value >0.20. Dietary pattern scores were categorized into quartiles for further analysis.

We compared characteristics between EGWG and non-EGWG pregnancies using Chi-square tests and compared pattern scores across different social demographic variables using t-tests or ANOVA. We estimated the odds ratio (OR) for EGWG in relation to the dietary pattern score (in quartiles) using logistic regression models. We reported p-for-trend to test for linear trends in multivariable models by using the median score of each dietary pattern quartile as a continuous variable. We conducted unadjusted and adjusted models in the analysis, as follows: Model 1, unadjusted (individual dietary pattern); Model 2: adjusted for other dietary pattern scores; Model 3: Model 2 plus age, race, education attainment, household income per year, parity, smoking status during pregnancy, physical activity status, and total energy intake per day. In the sensitivity analysis, we conducted linear regression models to examine the associations of the GWG rate with dietary pattern score quartiles and further adjusted women’s pre-pregnancy BMI. We further examined the potential effect modification by women’s pre-pregnancy weight status (non-OwOb vs. OwOb) by including multiplicative interaction terms in the models. 

All analyses were performed using Stata S.E. 16 (Stata Corp, College Station, TX, USA).

## 3. Results

### 3.1. Participant Characteristics

In our study, 52.5% of participants were reported to be EGWG (539 of 1026 participants). Compared with the participants included in our study (*n* = 1026), participants who were excluded (*n* = 270) were older, were less likely to be of Han ethnicity, more likely to be smokers, and had a lower physical activity level (Appendix A). Compared to women who did not have EGWG, women with EGWG were more likely to be a minority (18.9% vs. 14.2%), tended to have lower levels of education (rate of educational attainment of college or above: 72.5% vs. 80.9%) and were more likely to be OwOb (24.1% vs. 12.7%). No significant differences were observed between women with and without EGWG with respect to age, household income, parity, smoking status, energy intake, and physical activity status (Table 1).

### 3.2. Dietary Patterns

Four dietary patterns were identified, accounting for 27.1% of the total variation in the present analysis (Table 2). We designated the four dietary patterns according to food group factor loadings as follows: traditional pattern; sweet foods pattern; high protein pattern and milk–nut–seafood pattern. The traditional pattern consisted of high intakes of tubers (0.67), vegetables (0.61), fruits (0.55), red meat (0.54), and rice (0.42). The sweet foods pattern consisted of high intakes of sweet beverages (0.73), pastry and candy (0.65), shrimps, crabs and mussels (0.42), and fruits (0.24). The high protein pattern consisted of high intakes of fried foods (0.73), beans and bean products (0.69), dairy products (0.32), and fruits (0.21). The milk–nut–seafood pattern consisted of high intakes of milk (0.68), nuts (0.49), shrimps, crabs and mussels (0.39), fruits (0.27), dairy products (0.24), eggs and egg products (0.22), pastry and candy (0.20), and a lower intake of sweet beverages (−0.24).

Dietary pattern scores were positively associated with the daily energy intake level among all four patterns. In addition, women with higher traditional pattern scores were younger and with a lower level of education. Women who had higher sweet foods pattern scores had a higher household income per year (Table 3).

### 3.3. Dietary Patterns and Risk of EGWG

We assessed the associations between EGWG risk and dietary pattern scores (Table 4). In the full-adjusted models, we found that women in the third quartile (Q3) of the traditional pattern had a higher probability of EGWG (quartile 3 [Q3] vs. quartile 1 [Q1]: OR = 1.62, 95% CI = 1.10−2.38; *p* for trend = 0.03). Women in the second quartile (Q2) of the sweet foods pattern had a lower probability of EGWG (Q2 vs. Q1: OR = 0.70, 95% CI = 0.49−0.99). Women in the second and third quartile of the high protein pattern had a lower probability of EGWG (Q2 vs. Q1: OR = 0.68, 95% CI = 0.47−0.97; Q3 vs. Q1: OR = 0.56, 95% CI = 0.39−0.81). There were no significant associations between the milk–nut–seafood pattern and EGWG risk. Based on the sensitivity analysis further adjusted for pre-pregnancy BMI did not change the results, except that the association between the highest quartile of the traditional pattern (Q4 vs. Q1: OR = 1.67, 95% CI = 1.04−2.67) and high protein pattern (Q4 vs. Q1: OR = 0.67, 95% CI = 0.64−0.99) with a probability of EGWG became significant (Appendix A).

### 3.4. Dietary Patterns in Relation to EGWG, Stratified by Pre-Pregnancy Weight Status

In the stratified analyses (Figure 1), we showed that the associations between dietary pattern with risk of EGWG varied with pre-pregnancy weight (non-OwOb vs. OwOb). The positive association between traditional pattern and EGWG was more pronounced among OwOb women before pregnancy (*p* for interaction < 0.01). Compared with the quartile 1, the adjusted OR of the quartile 4 was 5.17 (95% CI = 1.45−18.46) for OwOb women and 1.35 (95% CI = 0.81−2.26) for non-OwOb women before pregnancy. Similarly, the protective influence of the high protein pattern on EGWG was more pronounced among non-OwOb women before pregnancy (*p* for interaction < 0.01). Compared with the quartile 1, the adjusted OR of quartile 3 was 0.54 (95% CI = 0.36−0.81) for non-OwOb women and 0.68 (95% CI = 0.23−2.02) for OwOb women before pregnancy.

### 3.5. Dietary Patterns in Relation to GWG Rate

Higher high protein pattern quartiles were also associated a lower rate of GWG (Q4 vs. Q1: β = −0.03 kg/week, 95% CI = −0.06 to −0.01; *p* for trend = 0.04). Other dietary patterns were not associated with the GWG rate (Appendix A).

## 4. Discussion

This is the first study to examine maternal dietary patterns in relation to risk of EGWG in northern China. We found that maternal adherence to a traditional pattern during pregnancy (higher intake of tubers, vegetables, fruits, red meat, and rice) was significantly associated with an increased risk of EGWG; the association was more pronounced among women who were overweight/obese before pregnancy. Maternal adherence to a high protein pattern (higher intake of fried foods, beans and bean products, dairy products, and fruits) was significantly associated with a decreased risk of EGWG; the association was more pronounced among women who were non-overweight/obese before pregnancy.

Our findings are in line with the findings of previous studies [9,12,13,14,18,19,28] that indicated prenatal dietary patterns have important roles in GWG status. We reported that women who adhered to a traditional pattern consisting of high intakes were at a higher risk of EGWG. A previous study among southern Chinese women also reported that maternal adherence to dietary patterns richer in fruits during pregnancy was associated with a higher GWG [18]. Two other studies indicated that dietary patterns with a higher intake of red meat were related to a higher probability of EGWG, which partly supported our study findings [12,14]. Possible mechanisms underlying the association between dietary pattern and EGWG could be that the over-consumption of high glycemic index foods, such as tubers, rice, and fruits and high energy dense foods, such as red meat, may influence maternal blood glucose levels and lead to excess fat storage among pregnant women. Even though the traditional pattern also contains food items, such as vegetables, they have been reported to have a beneficial effect on women’s health [13,19]. This common Chinese dietary pattern has an adverse influence on EGWG prevention.

In our study, higher adherence to a high protein pattern was related to a decreased risk of EGWG. The pattern was highly characteristic of protein-rich foods, such as beans, bean products, and dairy products. Even though fried foods are known as risk markers for the development of obesity, fried foods also contain an abundance of high-protein foods, such as fried chicken and fish. Previous studies have reported that dietary patterns containing protein-rich foods, such as beans [13,19], dairy [9,28], and fish [28], are associated with a lower risk of EGWG. Potential mechanisms could be that most of these high-protein foods, such as milk, bean products, and fish, are relatively low-energy dense foods compared to high-carbohydrate or -fat foods [29]. In addition, protein absorption consumes more energy than carbohydrates and fat, which leads to less energy storage [30,31]. High protein foods also provide a higher level of satiety [29] by increasing satiety-inducing hormones, such as peptide YY, glucagon-like peptide-1, and inhibiting hunger hormones [32,33], thus leading to reduced food intake among pregnant women.

The stratified influence of pre-pregnancy weight status on the relationship between maternal dietary patterns and GWG was less-studied. Our findings indicated that the association between traditional patterns on EGWG was more pronounced among women who were OwOb before pregnancy. An Italian cohort study reported similar results, although the sample size of that study was limited (*n* = 232) [12]. Specifically, adherence to an unhealthy Western dietary pattern was associated with increased GWG, especially among obese women. OwOb women have higher blood glucose levels than non-OwOb women [34], and thus may be more sensitive to the high-glycemic dietary patterns, such as traditional Chinese or Western patterns during pregnancy. We also observed that the protective association between high protein pattern were more pronounced among women who were non-OwOb before pregnancy, which is consistent with a previous study in South Africa [18]. Dietary patterns consisting of a higher intake of legumes and meats were associated with a decreased probability of EGWG among non-OwOb women, but not among OwOb women [16]. These findings highlighted the independent risk effect of pre-pregnancy obesity on EGWG risk. Despite that, according to our findings, a healthy diet during pregnancy could still help OwOb women reduce the probability of EGWG, although not significantly in the present study. Further studies are needed to explore beneficial dietary patterns to prevent EGWG among OwOb women.

The strengths of our study included the prospectively community-based study design, relatively large sample size, and that the study was conducted among the less-studied northern Chinese population. Our study had several limitations. First, despite the multicenter study design, the samples in the present study were regional populations, located in Shenyang, northern China, and thus may not be representative of other populations. Second, the study population was urban-based, and has a relatively high socioeconomic status, which could explain the high incidence of EGWG in our study. Third, dietary assessments were conducted during the second trimester of pregnancy, thus may not reflect the first and the third trimester diet status, though previous studies indicated that dietary patterns are likely to remain stable across pregnancy [35,36]. Fourth, the total variation of the four dietary patterns was relatively small (27.1%) in the present analysis, which is common in Chinese dietary pattern analysis. One of the possible reasons could be due to the complexity of Chinese dietary and other food intake combinations which were not identified as distinct dietary patterns. Fifth, even though we have provided a visual aid book to help the participants identify the portion size of foods, the self-reported food diaries may still lead to variability when recording the dietary patterns and the size of portions. Finally, there may be residual confounding factors that have not been considered in this study, such as genetic risk, which is associated with the risk of EGWG and pre-pregnancy status.

## 5. Conclusions

In summary, we identified an at-risk association between traditional patterns for pregnant women and EGWG, which consisted of high intakes of tubers, vegetables, fruits, red meat, and rice. The association was more pronounced among women who were OwOb before pregnancy. A protective high protein pattern, which consisted of high intakes of fried foods, beans and bean products, dairy products, and fruits was associated with a lower risk of EGWG, and the association was more pronounced among women who were non-ObOw before pregnancy. These findings could be helpful in the development of dietary guidelines during pregnancy to prevent EGWG in China.

## Figures and Tables

**Figure 1 nutrients-14-02551-f001:**
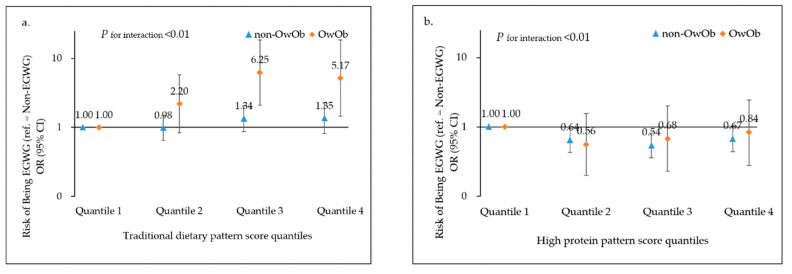
Associations between (**a**) traditional dietary pattern, (**b**) high protein dietary pattern score quantiles with risk of EGWG, stratified by pre-pregnancy weight (non-OwOb vs. OwOb). Adjusted for other dietary pattern scores, age, parity, family income, education level, ethnicity, smoking status, total energy intake per day, and physical activity per week. EGWG: excess gestational weight gain; OR: odd ratio; CI: confidence interval.

**Table 1 nutrients-14-02551-t001:** Maternal characteristics by gestational weight gain status (*n* = 1026).

Characteristics	Gestational Weight Gain Status, *n* (%)		
	Non-EGWG	EGWG	*p* Value
Age at enrollments (years)			0.26
<25	27 (5.5)	33 (6.1)	
25−29	220 (45.2)	230 (42.7)	
30−34	157 (32.2)	201 (37.3)	
≥35	83 (17.0)	75 (13.9)	
Ethnicity			0.04
Han	418 (85.8)	437 (81.1)	
Minority	69 (14.2)	102 (18.9)	
Educational attainment			<0.01
High school or below	93 (19.1)	148 (27.5)	
College or above	394 (80.9)	391 (72.5)	
Household income per year, CNY			0.37
<50,000	251 (51.5)	293 (54.4)	
≥50,000	236 (48.5)	246 (45.6)	
Parity			0.27
Primipara	370 (76.0)	425 (78.9)	
Multipara	117 (24.0)	114 (21.2)	
Pre-pregnancy BMI category, kg/m^2^			<0.01
<18.5	86 (17.7)	48 (8.9)	
18.5–<25.0	339 (69.6)	361 (67.0)	
≥25.0	62 (12.7)	130 (24.1)	
Smoking during or before pregnancy			0.49
No	485 (99.6)	535 (99.3)	
Yes	2 (0.4)	4 (0.7)	
Physical activity status during pregnancy, MET-hour/week			0.94
<100	125 (25.8)	141 (26.2)	
100−200	263 (54.0)	293 (54.4)	
>200	99 (20.3)	105 (19.5)	
Energy intake, kcal/day			0.15
<2100	305 (62.6)	314 (58.2)	
≥2100	182 (37.4)	225 (41.7)	

EGWG: excess gestational weight gain, CNY: Chinese Yuan, BMI: body mass index, MET: metabolic equivalent.

**Table 2 nutrients-14-02551-t002:** Factor loadings of derived dietary patterns from 3-day food diaries (*n* = 1026).

Dietary Patterns	Food	Factor Loading Coefficient	Variance Explained (%)
Traditional pattern			8.68
	Tubers	0.67	
	Vegetables	0.61	
	Fruits	0.55	
	Red meat	0.54	
	Rice	0.42	
Sweet foods pattern			6.39
	Sweet beverages	0.73	
	Pastry and candy	0.65	
	Shrimps, crabs and mussels	0.42	
	Fruits	0.24	
High protein pattern			6.14
	Fried foods	0.73	
	Beans and bean products	0.69	
	Dairy products	0.32	
	Fruits	0.21	
Milk-nut--seafood pattern			5.85
	Milk	0.68	
	Nuts	0.49	
	Shrimps, crabs and mussels	0.39	
	Fruits	0.27	
	Dairy products	0.24	
	Eggs and egg products	0.22	
	Pastry and candy	0.20	
	Sweet beverages	−0.24	

**Table 3 nutrients-14-02551-t003:** Dietary pattern scores according to participant characteristics (*n* = 1026).

Characteristics	Dietary Pattern Scores, Mean (SD)			
	Traditional Pattern	Sweet Foods Pattern	High Protein Pattern	Milk-Nut-Sea Food Pattern
Age, years				
<25	0.53 (1.89)	0.27 (1.67)	0.00 (1.80)	−0.32 (1.05)
25−29	0.07 (1.89)	0.01 (1.39)	0.00 (1.26)	0.06 (1.28)
30−34	−0.17 (1.74)	−0.01 (1.22)	0.04 (1.29)	−0.06 (1.18)
≥35	−0.01 (2.10)	−0.12 (1.38)	−0.09 (1.25)	0.09 (1.35)
*P*	0.040	0.298	0.756	0.092
Ethnicity				
Han	−0.01 (1.87)	0.00 (1.35)	0.02 (1.34)	0.02 (1.27)
Minority	0.07 (1.91)	0.01 (1.34)	−0.11 (1.11)	−0.09 (1.13)
*P*	0.598	0.944	0.208	0.312
Educational attainment				
High school or below	0.29 (1.92)	−0.14 (1.24)	−0.11 (1.21)	−0.06 (1.16)
College or above	−0.09 (1.86)	0.04 (1.38)	0.04 (1.33)	0.02 (1.27)
*P*	0.006	0.062	0.119	0.417
Household income per year, CNY				
<50,000	0.08 (1.92)	−0.08 (1.21)	−0.06 (1.28)	−0.07 (1.19)
≥50,000	−0.09 (1.83)	0.09 (1.49)	0.06 (1.33)	0.08 (1.30)
*P*	0.151	0.041	0.149	0.068
Parity				
0	0.00 (1.86)	0.02 (1.40)	0.02 (1.32)	0.03 (1.28)
≥1	−0.02 (1.95)	−0.08 (1.17)	−0.06 (1.25)	−0.12 (1.12)
*P*	0.887	0.328	0.404	0.104
Smoking status during pregnancy				
Yes	−0.75 (2.05)	0.03 (1.96)	−0.43 (1.04)	−0.41 (0.75)
No	0.00 (1.88)	−0.00 (1.35)	0.00 (1.30)	0.00 (1.25)
*P*	0.327	0.950	0.417	0.415
Pre-pregnancy BMI category, kg/m^2^				
<18.5	0.00 (1.87)	−0.02 (1.17)	−0.05 (1.27)	−0.06 (1.27)
18.5 to <25.0	0.02 (1.88)	0.01 (1.42)	−0.01 (1.31)	0.02 (1.24)
≥25.0	−0.08 (1.88)	−0.02 (1.22)	0.07 (1.30)	−0.05 (1.26)
*P*	0.778	0.944	0.669	0.674
Physical Activity, MET-hour/week				
<100	−0.02 (1.89)	0.12 (1.70)	−0.01 (1.43)	−0.10 (1.38)
100 to <200	0.02 (1.82)	−0.06 (1.13)	0.05 (1.27)	0.07 (1.20)
≥200	−0.02 (2.02)	0.00 (1.38)	−0.13 (1.21)	−0.07 (1.19)
*P*	0.944	0.205	0.218	0.129
Energy intake, kcal/day				
<2100	−0.86 (1.23)	−0.16 (1.10)	−0.25 (1.11)	−0.24 (1.07)
≥2100	1.30 (1.94)	0.24 (1.63)	0.39 (1.47)	0.37 (1.41)
*P*	<0.001	<0.001	<0.001	<0.001

CNY: Chinese Yuan, BMI: body mass index, MET: metabolic equivalent.

**Table 4 nutrients-14-02551-t004:** Crude and adjusted odds ratios for being EGWG by the quartiles of dietary pattern scores.

Dietary Patterns	Risk of Being EGWG (ref. = Non-EGWG, *n* = 1026)				
	Q1 Reference	Q2 OR (95%CI)	Q3 OR (95%CI)	Q4 OR (95%CI)	*P* _for trend_
Traditional pattern					
Model 1	1.00	0.98 (0.69, 1.38)	1.53 (1.08, 2.16)	1.52 (1.07, 2.15)	<0.01
Model 2	1.00	1.02 (0.71, 1.45)	1.62 (1.13, 2.31)	1.56 (1.08, 2.23)	<0.01
Model 3	1.00	1.06 (0.74, 1.53)	1.62 (1.10, 2.38)	1.57 (0.99, 2.50)	0.03
Sweet foods pattern					
Model 1	1.00	0.70 (0.50, 1.00)	0.95 (0.67, 1.34)	0.85 (0.60, 1.21)	0.81
Model 2	1.00	0.72 (0.50, 1.02)	1.04 (0.73, 1.50)	0.92 (0.64, 1.32)	1.00
Model 3	1.00	0.70 (0.49, 0.99)	1.05 (0.73, 1.51)	0.95 (0.66, 1.37)	0.86
High protein pattern					
Model 1	1.00	0.70 (0.49, 0.99)	0.61 (0.43, 0.86)	0.78 (0.55, 1.10)	0.23
Model 2	1.00	0.70 (0.49, 0.99)	0.57 (0.40, 0.82)	0.69 (0.48, 1.00)	0.12
Model 3	1.00	0.68 (0.47, 0.97)	0.56 (0.39, 0.81)	0.71 (0.48, 1.03)	0.16
Milk–nut–seafood pattern					
Model 1	1.00	0.85 (0.60, 1.20)	0.96 (0.68, 1.36)	1.08 (0.76, 1.53)	0.48
Model 2	1.00	0.84 (0.59, 1.20)	0.96 (0.68, 1.37)	1.01 (0.71, 1.46)	0.68
Model 3	1.00	0.85 (0.59, 1.22)	0.95 (0.66, 1.38)	1.05 (0.71, 1.54)	0.90

Model 1: Crude model. Model 2: Adjusted for other dietary pattern scores. Model 3: Model 2 + age, parity, family income, education level, ethnicity, smoking status, total energy intake per day, and physical activity status per week. EGWG: excess gestational weight gain, Q1: quartile 1; Q2: quartile 2; Q3: quartile 3; Q4: quartile 4; OR: odd ratio; CI: confidence interval.

## Data Availability

Data presented in this study are available on request from the corresponding author.

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
