# Peer review of "The Association between Dietary Patterns and Pre-Pregnancy BMI with Gestational Weight Gain: The “Born in Shenyang” Cohort"

_nutrients, 2022, doi:10.3390/nu14122551_

Round 1

Reviewer 1 Report

According to The Institute of Medicine and National Research Council (2009), four categories devided by prepregnancy BMI are underweight (<18.5), normal weight (18.5-24.9), overweight (25.0-29.9) and obese (>30). In this study, the authors defined normal prepregnant BMI as 18.5-24.0, and obesity as > 24.0. These defintions are not compatible with the Institute of Medicine categories. The authors  should explain the reasons why the definition has been changed or if possible, the authors should employ original categories. 

Reviewer 2 Report

Review: Associations of dietary patterns and pre-pregnancy BMI with gestational weight gain the
“Born in Shenyang” Cohort

Please consider changing the title to something like this:

The association between dietary patterns and pre-pregnancy BMI with gestational weight gain: the “Born in Shenyang” Cohort

Table 1 shows that educational attainment was significantly associated with EGWG, those who are more educated being more at risk. Could you add a sentence or two in the discussion on this please?

One of the study limitations could be the variability in the recording the dietary patterns and the size of portions. These were subject based not observer observed and could be another confounder.

Consider reducing the number of references. For example, no.7 to no. 20 report the same findings. You could choose the most relevant references to retain.

Line
Comment
132
  ...recommended
192
  Milk-nut-seaf ood = Milk-nut-sea food
276
  ...containing
279
  Replace “have” with “are”
282
  In addition, this is a repetition. Would you start the sentence with “High protein ...”
295
the protective” change to “that the protective”
318
combinations were” change to “combinations which were
